Review

Subject Area:
microbiology/genetics/biochemistry/structural biology

Keywords:
nucleoid, bacterial chromatin, horizontal gene transfer, environmental sensing

Author for correspondence:
R. T. Dame
e-mail: rtdame@chem.leidenuniv.nl

†These authors contributed equally to this study.

# The architects of bacterial DNA bridges: a structurally and functionally conserved family of proteins

L. Qin[1,2,†], A. M. Erkelens[1,2,†], F. Ben Bdira[1,2] and R. T. Dame[1,2]

[1]Leiden Institute of Chemistry, Leiden University, Einsteinweg 55, 2333CC Leiden, The Netherlands
[2]Centre for Microbial Cell Biology, Leiden University, Leiden, The Netherlands

 RTD, 0000-0001-9863-1692

Every organism across the tree of life compacts and organizes its genome with architectural chromatin proteins. While eukaryotes and archaea express histone proteins, the organization of bacterial chromosomes is dependent on nucleoid-associated proteins. In *Escherichia coli* and other proteobacteria, the histone-like nucleoid structuring protein (H-NS) acts as a global genome organizer and gene regulator. Functional analogues of H-NS have been found in other bacterial species: MvaT in *Pseudomonas* species, Lsr2 in actinomycetes and Rok in *Bacillus* species. These proteins complement *hns⁻* phenotypes and have similar DNA-binding properties, despite their lack of sequence homology. In this review, we focus on the structural and functional characteristics of these four architectural proteins. They are able to bridge DNA duplexes, which is key to genome compaction, gene regulation and their response to changing conditions in the environment. Structurally the domain organization and charge distribution of these proteins are conserved, which we suggest is at the basis of their conserved environment responsive behaviour. These observations could be used to find and validate new members of this protein family and to predict their response to environmental changes.

## 1. Introduction

All organisms compact and organize their genomic DNA. Structuring of the genome is achieved by the action of small, basic architectural proteins that interact with DNA. These proteins wrap, bend and bridge DNA duplexes. Despite the lack of both sequence and structural homology between architectural proteins in species across the tree of life, the basic concepts appear conserved, with all organisms harbouring functional analogues [1]. An essential feature of genome organization is its intrinsic coupling to genome transactions, such that a process like a gene expression is both dependents upon chromatin structure and a driving factor in chromatin (re)organization [2]. The structure of chromatin is affected by environmental signals, which can be translated into altered gene expression [3].

The best known architectural proteins are the histones expressed by eukaryotes. Binding of these proteins yields nucleosomes in which DNA is wrapped around an octameric histone protein core. Aided by other architectural proteins these nucleosomal fibres are further organized into higher-order structures [1,4–6]. Histone H1 and BAF (barrier-to-autointegration factor) are examples of eukaryotic architectural proteins that are capable of bridging DNA [7,8]. In addition, structural maintenance of chromosome (SMC) proteins (e.g. cohesin and condensin) act upon chromatin, forming large chromatin loops by bridging, at the expense of ATP [9]. SMC proteins are the only chromatin proteins that are universally conserved [10,11]. Finally, eukaryotes express small proteins that bend DNA, such as HMG-box proteins [12]. Archaea also express histones. Different from their eukaryotic counterparts, archaeal histones assemble into oligomeric

filaments along DNA, yielding hypernucleosomes [13–16]. In addition, they express DNA-bridging proteins such as Alba (acetylation lowers binding affinity), which can both form nucleofilaments and bridge DNA, depending on the protein : DNA stoichiometry [17,18]. Some archaeal species lack histones and express DNA-bending proteins instead, like Cren7 and Sul7 [19].

Bacteria lack homologues of the histone proteins expressed by eukaryotes and archaea. The organization of bacterial genomes is dependent on a group of architectural proteins collectively referred to as nucleoid-associated proteins (NAPs). At least 12 NAPs have been described for *Escherichia coli* and closely related species [20–22]. A shared feature of many of these proteins is their ability to bend DNA. Examples include the histone-like protein from strain U93 (HU), integration host factor (IHF) and the factor for inversion stimulation (Fis) [23–25]. The histone-like nucleoid structuring protein (H-NS) has an overarching role in the organization of the *E. coli* genome and acts as a global regulator of gene expression: 5–10% of *E. coli* genes are affected, mostly repressed, by H-NS [26]. Due to its preference for A/T-rich DNA, it specifically targets and silences horizontally acquired genes, a process referred to as xenogeneic silencing [27]. Key to the role of H-NS in both processes is the formation of nucleofilaments along the DNA and protein-mediated DNA–DNA bridges [28–30]. H-NS-like proteins are passive DNA bridgers in contrast with SMC proteins which are active, ATP-driven DNA bridgers (figure 1).

Over the last two decades, functional homologues of H-NS have been identified in other bacterial species. Despite low sequence similarity, these proteins have similar DNA-binding properties, resulting in the formation of structurally and functionally similar protein–DNA complexes. This ability is elegantly demonstrated by the genetic complementation of *hns⁻* phenotypes (like mucoidy, motility and β-glucoside utilization) in *E. coli* by MvaT from *Pseudomonas* species and Lsr2 from *Mycobacterium* and related actinomycetes [31,32]. *In vitro* both proteins are also able to bridge DNA in a manner similar to H-NS [28,33,34] (figure 1). MvaT regulates hundreds of genes in *Pseudomonas aeruginosa* and Lsr2 binds to one fifth of the *Mycobacterium tuberculosis* genome, especially to horizontally acquired genes [35–38]. These properties endow them with functions as global gene regulators and spatial chromatin organizers. A newly proposed functional homologue of H-NS is the repressor of *comK* protein (Rok) of *Bacillus subtilis*. This classification is primarily based on the observation that Rok binds extended regions of the *B. subtilis* genome and especially A/T rich regions acquired by horizontal gene transfer, which it aids to repress [39]. This specific property of silencing foreign genes makes Rok, just like H-NS, MvaT and Lsr2, a xenogeneic silencer. It is also associated with a large subset of chromosomal domain boundaries identified in *B. subtilis* by Hi-C [40]. As such boundaries might involve genome loop formation, this could imply a role as DNA-bridging architectural protein.

In this review, we focus on the properties of DNA-bridging proteins in bacteria with a proposed role in genome architecture and gene regulation: H-NS, MvaT, Lsr2 and Rok. We describe and compare their structure and function to define conserved features. Also, we discuss the mechanisms by which the architectural and regulatory properties of these proteins are modulated.

royalsocietypublishing.org/journal/rsob    Open Biol. 9: 190223

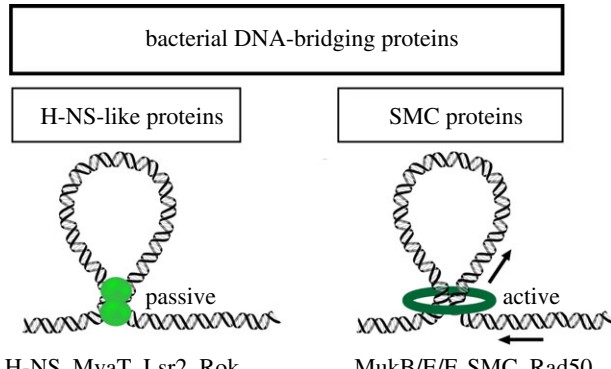

**Figure 1.** Bacterial DNA-bridging proteins. Two types can be distinguished: passive DNA bridgers such as H-NS-like proteins (light green), which bind distant segments of DNA duplexes and bring them together, and active DNA bridgers such as SMC proteins (dark green), which are able to connect two double stranded DNA segments, translocating along the DNA molecule with motor activity resulting from ATP hydrolysis. Note that the exact molecular mechanisms by which SMC proteins operate and are involved in loop formation only start to be defined and are a topic of much discussion.

## 2. Fold topology of H-NS-like proteins

Structural studies have revealed that H-NS, Lsr2 and MvaT harbour similar functional modules: (i) an N-terminal oligomerization domain consisting of two dimerization sites, (ii) a C-terminal DNA-binding domain and (iii) an unstructured linker region (figure 2*a–c*) [41–43,45,46]. For Rok, a similar overall domain architecture has been found: the C-terminal domain is capable of DNA binding and the N-terminal domain is responsible for oligomerization (figure 2*d*) [44].

### 2.1. The N-terminal domain

The N-terminal domain of H-NS and MvaT is involved in the formation of oligomers, which is a property essential for gene repression [37,47]. Both Lsr2 and Rok are capable of oligomerization, but it is currently unknown whether oligomerization is required for gene regulation [43,44]. As the N-terminal structure of most of these DNA-bridging proteins is known, differences and similarities in the mechanism of forming high-order complexes have become evident. H-NS of *Salmonella typhimurium* has two dimerization sites in the N-terminal domain (1–83) [41]. The N-terminal dimerization domain (site 1, 1–40) is formed by a 'hand-shake' topology between α1 and α2 and part of α3. The central dimerization domain (site 2, 57–83) has two α-helices α3 and α4 that form a helix–turn–helix dimerization interface. H-NS dimers are formed via site 1 in a tail-to-tail manner, which can oligomerize via site 2 via head-to-head association (figure 2*a*). The resulting crystal structure is superhelical. Therefore, it was proposed that DNA-H-NS-DNA filaments involve superhelical wrapping of DNA around an oligomeric protein core. However, apart from the X-ray crystal structure [41], there is no evidence for this type of H-NS nucleofilaments organization.

The crystal structure of the MvaT homologue, TurB from *Pseudomonas putida*, revealed a similar fold topology of the N-terminal dimerization site 2 as that of H-NS [42]. By contrast, site 1 exhibits a standard 'coiled-coil' architecture in MvaT/TurB, whereas H-NS due to the presence of two additional N-terminal helices (α1 and α2 in H-NS) compared with TurB/MvaT exhibits a 'hand-shake' topology (figure 2).

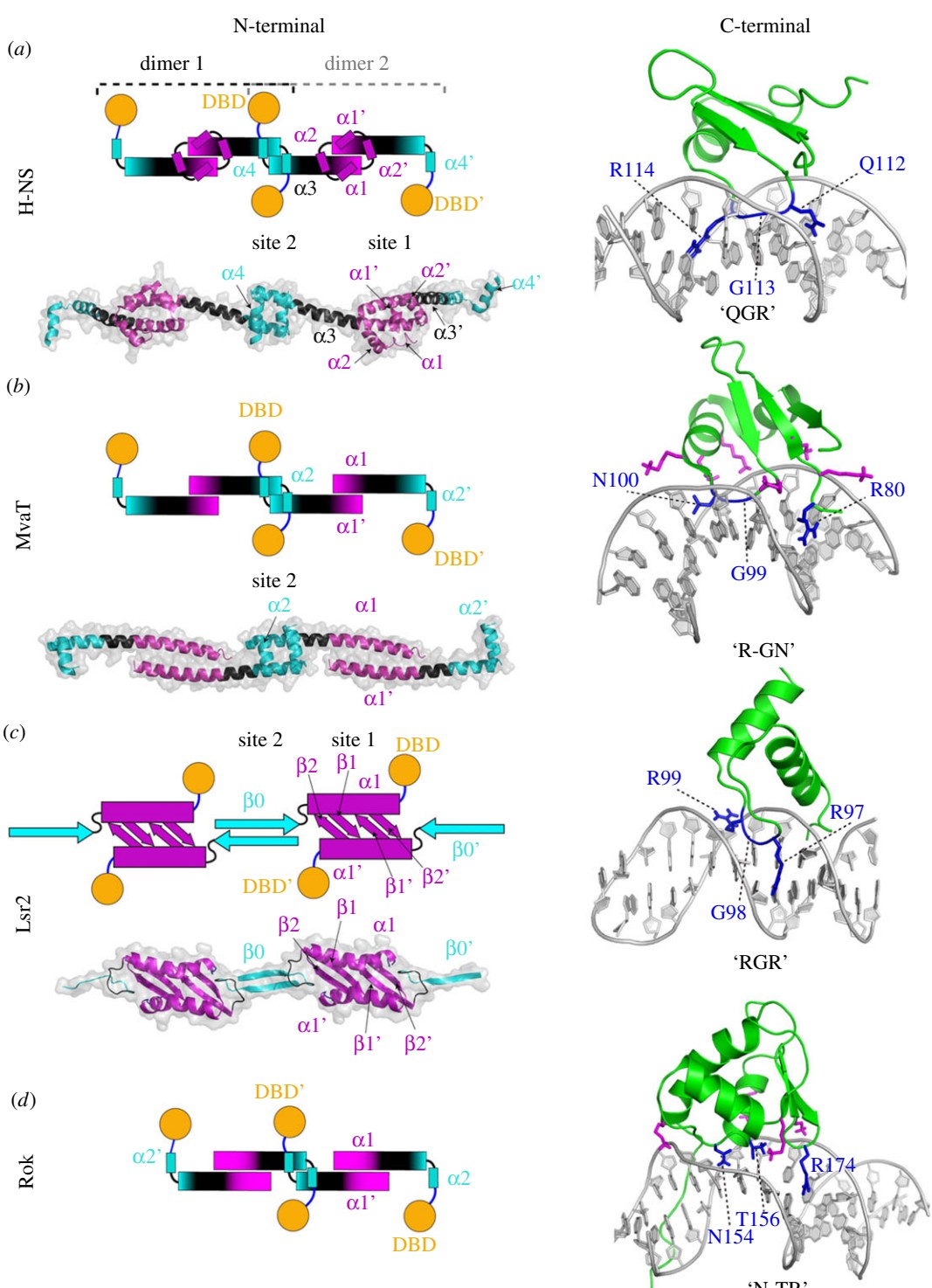

**Figure 2.** Fold topology and oligomerization states of H-NS-like proteins. Left panels show models of the structures of the N-terminal oligomerization domains of (*a*) H-NS [41], (*b*) MvaT [42] and (*c*) Lsr2 [43] as determined by crystallography. A schematic of the higher-order oligomerization states of H-NS-like proteins is shown above the crystal structures of the N-terminal domains. (*d*) For Rok, the schematic of its higher-order oligomerization state is based on secondary structure prediction. The dimerization sites (site 1) are shown in magenta and the oligomerization sites (site 2) in cyan. The DNA-binding domains are shown in orange spheres and the linker regions in blue lines. Right panels show the NMR structures of the DNA-binding domains of H-NS-like proteins and their DNA recognition mechanisms [44]. The loops of the DNA-binding motif are shown in blue and the residues involved in the direct interactions to the DNA minor groove and in the complex stabilization are shown in sticks.

Despite this difference in site 1, both proteins form a head–head and tail–tail dimer organization in their protein filaments.

The N-terminal structure of Lsr2 from *M. tuberculosis*, however, is completely different from that of H-NS and MvaT [43]. The flexible N-terminus is followed by a β-sheet formed by two antiparallel β-strands and a kinked α-helix. When forming dimers, the two β-sheets of the monomers align to form a four-stranded antiparallel β-sheet with an antiparallel arrangement of the α-helices on the opposite sides of the sheet (figure 2*c*). Notably, oligomerization does not occur with the first four amino acids of Lsr2 of *M. tuberculosis* present, but is triggered by trypsin cleavage, removing these residues [43]. The oligomerization between Lsr2 dimers occurs through an antiparallel association between two N-terminal β-strands

from adjacent monomers (figure 2*c*). The triggering of Lsr2 oligomerization by proteolysis indicates that this process is possibly controlled via protease activity *in vivo*, offering a mechanism for genome protection by Lsr2 under stress conditions [48]. Note, however, that these four amino acids are not highly conserved and are lacking in *Mycobacterium sinensis* Lsr2 (Genbank: AEF37887.1).

Although the three-dimensional structure of Rok's N-terminal domain has not been experimentally determined, it is predicted to contain two α-helices (α1 and α2) and a part of the first α-helix (residues 1–43) is predicted to form a 'coiled-coil' dimerization motif [49,50] (figure 2*d*), similar to MvaT. Based on these secondary structure predictions it is plausible that Rok exhibits higher-order oligomerization with a structural organization resembling that of TurB/MvaT.

## 2.2. The C-terminal domain

The C-terminal domain of all four proteins recognizes and binds to DNA. *In vivo*, the DNA segments bound are generally AT-rich compared to other parts of the genome. Although H-NS and Lsr2 differ in overall structure of the C-terminal domain, both proteins recognize the minor groove of DNA with a similar 'AT-hook-like' motif composed of three consecutive 'Q/RGR' residues [45] (figure 2*a*,*c*). H-NS and Lsr2 generally favour similar AT-rich DNA target sequences, both with a preference for TpA steps over A-tracts [45]. This can be related to the width of the minor groove. A-tracts narrow the minor groove in comparison to TpA steps, while GC-rich sequences result in a wider minor groove. TpA steps result in a favourable width for H-NS and Lsr2 binding to the DNA [45].

The C-terminal domain of MvaT exhibits a similar overall fold as H-NS, but has a different DNA-binding mechanism [46] (figure 2*b*). The C-terminus of MvaT recognizes AT-rich DNA via both the 'AT-pincer' motif consisting of three non-continuous residues 'R-G-N' targeting minor groove DNA, and a 'lysine network' interacting with the DNA backbone by multiple positive charges. In general, MvaT has similar preferences in binding DNA sequences as H-NS and Lsr2, preferring TpA steps over A-tracts [46]. MvaT is, however, more tolerant to G/C interruptions in the DNA sequence than H-NS and Lsr2.

Binding of the C-terminal domain of H-NS and Lsr2 to DNA causes no notable changes in DNA conformation [36,45], whereas the C-terminus of MvaT triggers significant distortions in the DNA molecule [46]. It is likely that the 'AT-hook motif' of H-NS and Lsr2 forms a narrow crescent-shaped structure that inserts into the minor groove without significantly interrupting the DNA helical trajectory. When bound by the C-terminal domain of MvaT, the minor groove of DNA is expanded leading to a significant rearrangement in base-stacking [46]. Therefore, binding of full-length MvaT dimers to DNA results in DNA bending [51].

The structure of the C-terminal DNA-binding domain of Rok reveals that it employs a winged helix domain fold using a unique DNA recognition mechanism different from the other three proteins [44] (figure 2*d*). Rather than using an 'AT-hook-like' motif or 'AT-pincer' motif, the C-terminus of Rok targets the DNA minor groove via the three non-continuous residues N-T-R. As in the case of MvaT, DNA binding is stabilized by a hydrogen bond network between

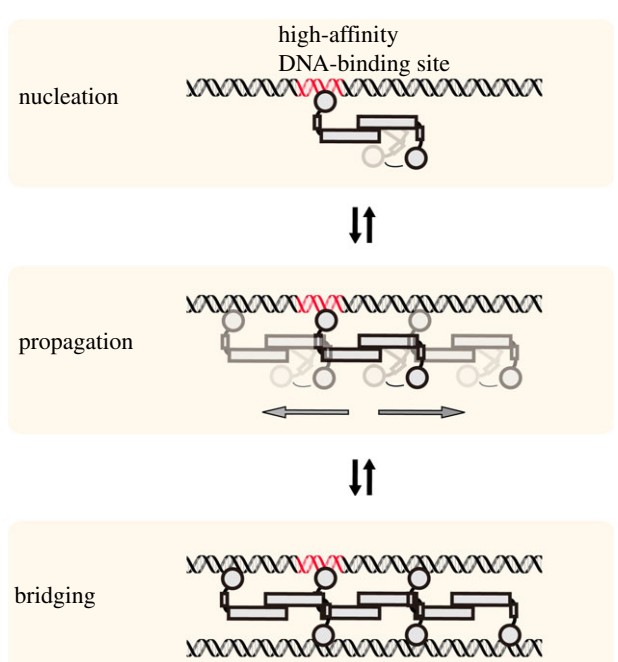

**Figure 3.** Assembly of functional protein–DNA complexes by H-NS-like proteins. DNA binding of H-NS-like proteins initiates at a nucleation site (high-affinity binding site, red) in the genome; H-NS-like proteins propagate along DNA in a cooperative way due to protein–protein interactions; DNA–protein–DNA-bridging complex can be formed under favourable bridging conditions, bringing distant DNA duplexes together. Further propagation (not indicated in the figure) may occur in the bridged complex due to both protein–protein interactions and high effective local DNA concentration. Note that all steps are reversible, which is important for modulation of the function of these proteins (§5).

several lysine residues and the phosphate groups of the DNA backbone [44]. Similar to H-NS, Lsr2 and MvaT, Rok interacts with AT-rich DNA sequences with a preference for TpA steps [44]. Rok has selectivity towards some specific DNA sites, comparable with the affinity of H-NS for its high-affinity sites, where the highest affinity is noted for AACTA and TACTA sequences [52]. Compared with MvaT, Rok induces a more pronounced conformational change in its target DNA substrate. Rok binding leads to bending of DNA by approximately 25° [44].

The function of the conformational changes in target DNA induced by MvaT and Rok binding is unknown, but the changes may be of importance in gene regulation, where often multiple architectural proteins operate in concert. An example of this is the reversion of Rok repression by ComK at the *comK* promotor [53]. It has been suggested that the DNA-bending by Comk reverses the conformational changes in the DNA induced by Rok.

# 3. Protein–DNA complexes formed by H-NS-like proteins

Two types of protein–DNA complexes can be formed by H-NS-like proteins: (i) nucleoprotein filaments and (ii) bridged complexes. Assembly of protein–DNA complexes by H-NS, MvaT and Lsr2 is believed to proceed via a multi-step process (figure 3). First, the C-terminal DNA-binding domain directs the protein to a high-affinity site (nucleation) [52]. This step

is probably assisted by the positively charged amino acid residues of the linker region, which interact with the DNA and recruit H-NS to bind non-specifically [54–56]. This then allows H-NS to scan on DNA to search for the specific site where the C-terminal domain can engage with a higher affinity. Next, the proteins spread cooperatively along the DNA forming a nucleoprotein filament by oligomerization through their N-terminal domains. If the surrounding conditions are favourable, these nucleoprotein filaments can interact with another DNA duplex to form a bridge. Both types of protein–DNA complexes are thought to play important roles in genome structuring and gene silencing. Evidence in support of formation of nucleoprotein filaments comes from atomic force microscopy and single-molecule studies which revealed that H-NS, Lsr2 and MvaT all form rigid protein–DNA filaments, suggestive of protein oligomerization along DNA [30,57–60]. DNA–DNA bridging has been visualized *in vitro* using microscopy [28,33,34] and corroborated using solution based assays [29,61]. The ability to oligomerize is important for the function of these proteins in chromosome organization and gene regulation [37,59,60]. To date, there are no indications that Rok rigidifies DNA suggesting that this protein might not be able to oligomerize along DNA. Nevertheless, the protein induces DNA compaction and is capable of DNA–DNA bridging [62].

Following the initial observations that two types of complexes can be formed [34,37,58,60,63], the mechanism that drives the switch between these two DNA-binding modes has long been elusive. Two recent studies on H-NS have provided the first mechanistic insights into this process [61,64]. The switch between the two DNA-binding modes [58] involves a conformational change of the H-NS dimers from a half-open to an open conformation driven by $Mg^{2+}$ [61]. These conformational changes are modulated by the interactions between the N-terminal domain of H-NS and its C-terminal DNA-binding domain. Mutagenesis at the interface of these domains generated an H-NS variant no longer sensitive to $Mg^{2+}$, which can form filaments and bridge DNA [61]. Recently, these interactions were confirmed by Arold and co-workers using H-NS truncated domains [64]. The linker of H-NS was shown to be essential for the interdomain interaction between the N-terminal and C-terminal domain [64]. Studies on MvaT in our laboratory further support a model in which both an increase in ionic strength and the DNA substrate additively destabilize these interdomain interactions, inducing the dimers to release their second DNA-binding domain to bind and bridge a second DNA molecule in trans (figure 4*a*) [51].

The interdomain interactions described above for H-NS and MvaT might be driven by the asymmetrical charge distribution within the protein sequence: the N-terminal domain is mainly negatively charged, while the linker and the DNA-binding motif are positively charged (figure 4*b*,*c*). Analysis of the average charge of the primary sequences of H-NS-like proteins revealed that this characteristic is a conserved feature among H-NS/MvaT proteins across species and extends to Lsr2 (figure 4*b*–*d*; electronic supplementary material) [51]. The conserved asymmetrical charge distribution might provide an explanation for how H-NS, MvaT and Lsr2 act as sensors of environmental changes. For Rok, this asymmetrical charge distribution between its folded domains is less pronounced (figure 4*e*). In addition to that, Rok contains a neutral Q linker instead of the basic linker

integrated into H-NS, MvaT and Lsr2 polypeptide chains. Previously, the Q linker was defined as a widespread structural element connecting distinct functional domains in bacterial regulatory proteins [65]. Thus, the difference in charge distribution and the linker region between Rok and the other proteins could have functional implications (see §5).

# 4. Functional properties

## 4.1. Genome organization

The chromosomal DNA of *E. coli* is structured into domains of various sizes [66]. The first layer of organization involves division in four macrodomains (Ori, Ter, Right and Left) of about 1 Mb in size [67–70]. Although it is not completely clear how the borders of these domains are formed, several *E. coli* DNA-binding proteins (e.g. SeqA, SlmA and MatP) are associated with certain macrodomains only [70–73]. One scale smaller, microdomains have been described as roughly 10 kb in size, which are attributed to loop formation in *E. coli* [74–77]. H-NS is important for domain formation *in vivo* and the distribution of H-NS along the chromosome is suggestive of a role in establishing microdomains [75,78]. Recently, it was shown with Hi-C that H-NS mediates short range contacts along the chromosome [79]. The DNA-bridging ability of H-NS matches well with the structural properties detected *in vivo*. Genome-wide 3C-based studies reveal chromosomal interaction domains (CIDs) along the chromosome in *Caulobacter crescentus*, *B. subtilis* and *Vibrio cholerae* [40,80,81]. These domains are tens to hundreds of kilobase pairs in size and the boundaries of CIDs are often formed by highly transcribed genes [80].

It is likely that the other DNA-bridging proteins organize the bacterial genome in similar ways. The genome of *B. subtilis* consists of three global domains and local smaller domains [40]. A subset of the barriers between local domains corresponds to genomic positions bound by Rok [39,40]. This suggests a role for Rok in the genome organization of *B. subtilis*. For *Pseudomonas* and *Mycobacterium* species, such studies have not been done yet. But ChIP-on-chip data shows that MvaT and Lsr2 bind to defined regions throughout the whole genome [35,36], in support of a similar role in genome organization as H-NS and Rok.

## 4.2. Gene regulation by H-NS-like proteins

There has been a lot of discussion in the field as to which DNA-binding mode is relevant *in vivo* and which of the two modes is needed for gene silencing. The short answer is that both modes of binding could explain gene repression. H-NS-DNA filament formation at or across a promotor region potentially occludes RNA polymerase (RNAP), preventing the initiation of gene transcription. H-NS mutants that are incapable of gene silencing were indeed found *in vitro* to be defective for nucleofilament formation [59]. Note, however, that oligomerization is essential to both filament formation and bridging, and also that transcription can be affected by the failed assembly of both types of complexes, if oligomerization is perturbed. Data from a lot of recent *in vitro* studies indeed favour models in which DNA-bridging plays a key role. Promotors that are sensitive to local DNA topology might be inactivated by H-NS due to

royalsocietypublishing.org/journal/rsob  Open Biol. **9**: 190223

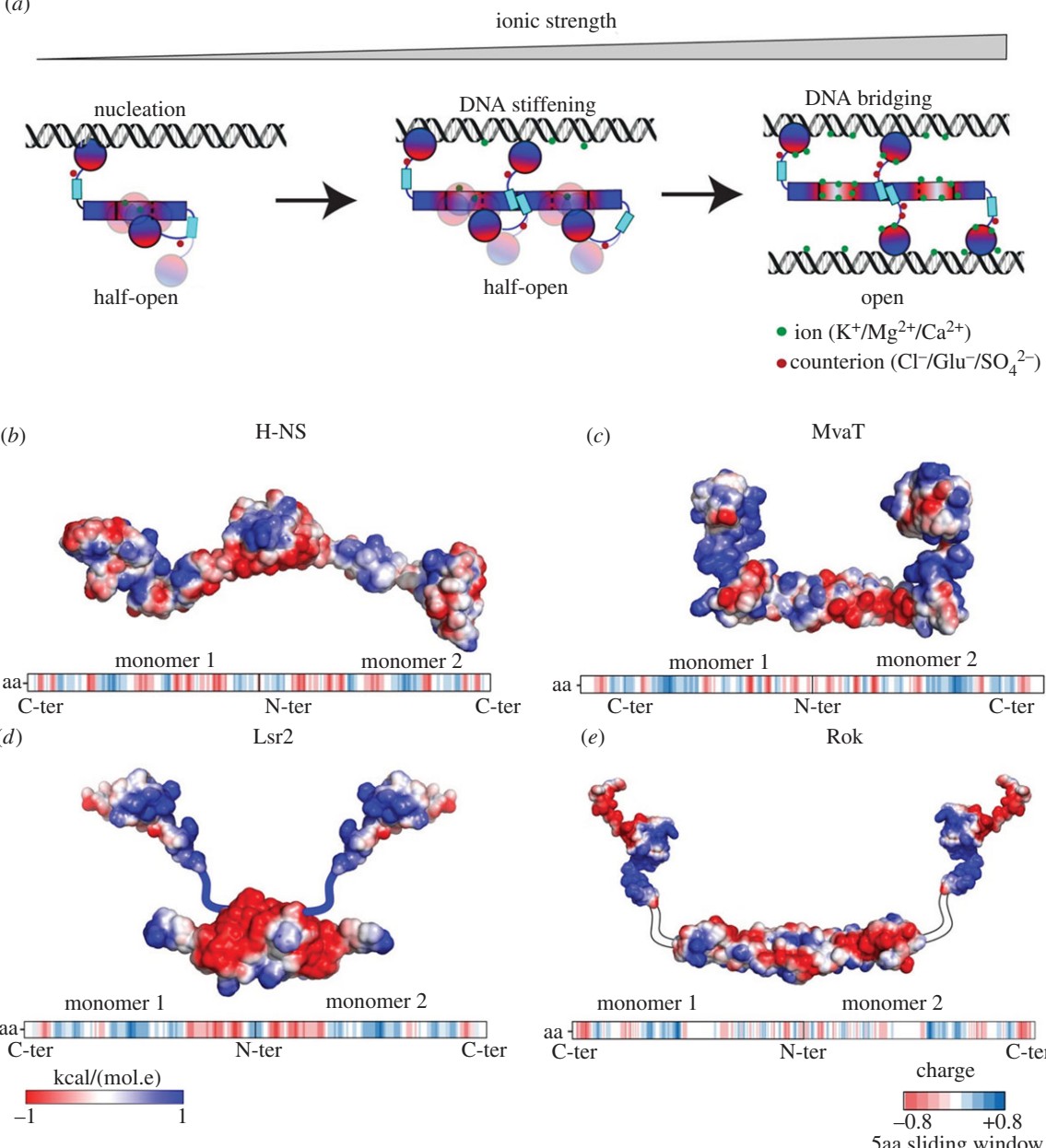

**Figure 4.** H-NS-like protein DNA-binding modes and electrostatics. (a) Schematic of the switching mechanism between H-NS-like proteins DNA-binding modes. The red/blue gradient represents the electrostatics of H-NS-like protein surfaces [51,61]. The red and blue are for negatively and positively charged surface regions, respectively. The electrostatic potential surfaces of (b) H-NS, (c) MvaT, (d) Lsr2 and (e) Rok are depicted on full-length protomer structural models of the proteins using a red/white/blue colour gradient. The missing linkers in the structural models of Lrs2 and Rok are shown in blue (positively charged) and white (neural), respectively. In the lower panel, the five amino acid sliding window averaged charge of the proteins protomers primary sequences generated by EMBOSS charge is shown. Positive, negative and neutral charged amino acid fragments are shown in blue, red and white bars, respectively.

its ability to constrain supercoils. [82] Generally, these promotors are affected at a distance and H-NS mediated bridging could also constrain supercoils by generation of a diffusion barrier. Bridge formation in promotor regions not only has the potential to occlude RNAP, but also to physically trap RNAP, thereby preventing promoter escape and silencing the associated gene [83–85]. Evidence from genome-wide binding profiles and a protein–protein interactions study indicates that H-NS and RNAP colocalize at some promotors, which is compatible with a trapping mechanism [86,87]. *In vitro* transcription studies show that while bridged H-NS-DNA complexes can inhibit progression of the elongating RNAP, the H-NS-DNA filament cannot, highlighting the importance of H-NS' DNA-bridging activity [88,89]. Further mechanistic dissection of the repressive role of H-NS in

transcription awaits single molecule *in vitro* transcription experiments and the application of novel approaches permitting structural investigation of the process *in vivo*. Much less mechanistic information is available for the other architectural DNA-bridging proteins. Lsr2 can inhibit transcription *in vitro* and inhibits topoisomerase I, thereby introducing supercoils into the DNA [38]. Rok reduces the binding of RNAP at the promotor of *comK* [53]. Rok is antagonized at this promotor by ComK itself and, although their binding sites partially overlap [90–92], this occurs without preventing Rok binding [53]. It was postulated that anti-repression is achieved through modulation of DNA topology, which would imply that Rok itself also has impact on DNA topology [53]. In *P. aeruginosa*, MvaT is known to repress the *cupA* gene, which is important in biofilm formation.

royalsocietypublishing.org/journal/rsob    Open Biol. 9: 190223

Mutants that were unable to silence *cupA* could not form a nucleofilament [57]. For these three proteins, it remains to be investigated if DNA bridging is as important for gene silencing as in the case of H-NS.

# 5. Functional regulation

## 5.1. Environmental conditions

Bacteria need to respond fast to environmental changes to be able to adapt to diverse living conditions. The bacterial genome operates as an information-processing machine, translating environmental cues into altered transcription of specific genes required for adaptation and survival [3]. Key to this process is the dynamic organization of the bacterial genome driven by such cues [2].

### 5.1.1. The role of temperature

Change in temperature is one of the many environmental changes encountered by bacteria that need altered gene expression for adaptation. H-NS regulates the expression of many genes responsive to changes in environment. In *E. coli*, H-NS controls more than 60% of the temperature-regulated genes: they have higher expression at 37°C than at 23°C [93–95]. These genes are involved in the nutrient, carbohydrate and iron utilization systems and their changes in expression at 37°C may be of advantage for host colonization [96–98]. The involvement of H-NS in temperature regulation is corroborated by *in vitro* transcription studies, which indicate that H-NS is no longer able to pause elongating RNAP at 37°C (see §4) [89]. The underlying mechanism may be an increased off-rate of H-NS, or a reduced propensity to oligomerize at the high temperature as observed *in vitro* [29,30,88,93].

Analogous to the situation for H-NS, also Lsr2–DNA filament formation is sensitive to temperature changes *in vitro*: the rigidity of Lsr2–DNA complexes is lower at 37°C than at 23°C [60]. This can be due to either a change in the ability of Lsr2 to oligomerize along DNA or a change in Lsr2–DNA-binding affinity at 37°C compared with 23°C. Qualitatively, Lsr2-induced DNA folding seems insensitive to change in temperature, suggesting but not conclusively proving that, DNA bridging by Lsr2 is not affected by temperature [60]. However, if the general binding scheme depicted in figure 3 applies, the effects of temperature on protein–DNA filament may translate into effects on DNA–protein–DNA-bridging activity. Thus, taken together, it might very well be that Lsr2 is involved in regulating thermosensitive genes in *Mycobacteria*.

Unlike the protein–DNA filaments formed by H-NS and Lsr2, the stiffness of MvaT–DNA filaments is not altered in the range of 23–37°C [57]. In the same temperature range, an increase in DNA compaction was observed with increasing temperature. Such a switch between bridging and non-bridging modes is suggestive of temperature-controlled gene regulation at MvaT-bridged genes in *Pseudomonas*.

DNA bridging by Rok is not sensitive to changes in temperature from 25°C to 37°C, suggesting that genes in *B. subtilis* that are repressed by Rok are not regulated by temperature [62].

### 5.1.2. The role of salt

H-NS has also been shown to regulate genes that are sensitive to salt stress. The *proU* (*proVWX*) operon is one of the best-studied operons that is osmoregulated by H-NS. The expression of *proU* is significantly upregulated by high osmolarity [99,100]. *In vitro*, the stiffness of the nucleoprotein filament formed by H-NS and Lsr2 is sensitive to change in salt concentration from 50 to 300 mM KCl [58,63]: the rigidity of the protein–DNA complexes decreased as salt concentration increased. However, the stiffness of the MvaT–DNA filament is not affected by salt over the same concentration range [57]. The formation of DNA–DNA bridges by H-NS and MvaT is sensitive to both $MgCl_2$ and KCl [51,58,101]. Changes in $MgCl_2$ (0–10 mM) or KCl concentration (50–300 mM) drive a switch between the DNA stiffening mode and bridging mode. This structural switch could be the mechanism underlying regulation of osmoregulated genes by H-NS and MvaT. Qualitatively, Lsr2-induced DNA folding seems insensitive to change in salt, suggesting that the formation of the Lsr2–DNA bridged complex is not affected by salt [60]. However, the salt effects on the structure of the Lsr2–DNA filament could alter the activity of DNA–Lsr2–DNA bridging. DNA bridging by Rok is independent of and not sensitive to both $MgCl_2$ (0–60 mM) and KCl (35–300 mM) concentration [62]. This insensitivity to changes in salt concentration may be related to the different charge distribution of Rok compared to the other H-NS-like proteins (figure 4), where the lack of charges may lead to less interdomain interactions. In this way, Rok could be always in an open conformation, suitable for DNA bridging.

### 5.1.3. The role of pH

H-NS has been reported to be involved pH-dependent gene regulation [102]. *In vitro*, the rigidity of H-NS-DNA nucleofilaments is shown to be sensitive to changes in pH [58]. A reduction in stiffness of the protein–DNA complex was observed with increasing pH from 6.5 to 8. The formation of the bridged DNA-H-NS-DNA complex might be insensitive to pH changes as H-NS induced DNA folding was unaffected over the same range of pH values [58].

Different from the pH-sensitivity observed for H-NS-DNA filaments, Lsr2–DNA nucleofilaments were shown to be insensitive to changes in pH from 6.8 to 8.8. Similar to H-NS, DNA folding by Lsr2 is not sensitive to pH changes in the same range [60].

Also the MvaT–DNA nucleoprotein filament is not sensitive to pH changes from 6.5 to 8.5, but DNA compaction is affected [57]: MvaT induced stronger folding at pH 6.5 than pH 8.5. Note that the DNA folding induced by H-NS, MvaT and Lsr2 was detected by a qualitative 'folding assay', in which observed DNA folding does not necessarily result from DNA bridging. A quantitative assay such as the 'bridging assay', developed by van der Valk *et al*. [103], is essential to better determine the sensitivity of DNA–protein–DNA bridging to changes in environmental conditions. With this bridging assay, DNA bridging by Rok was shown to be insensitive to pH changes from 6.0 to 10.0. Strikingly, even crossing the pI of Rok (9.31) did not affect its bridging capacities, indicating that charge interactions are unlikely to play a role in DNA bridging by Rok [62].

**Table 1.** Characteristics of bacterial DNA-bridging proteins.

| | H-NS | MvaT | Lsr2 | Rok |
|---|---|---|---|---|
| bacteria | *Enterobacteriaceae* (gram-negative) | *Pseudomonas* sp. (gram-negative) | *Actinomycetes* (gram-positive) | *Bacillus* sp. (gram-positive) |
| size (kDa) | 15.5 | 14.2 | 12.0 | 21.8 |
| protomer size | dimer | dimer | dimer | unknown |
| oligomerization | yes | yes | yes | yes |
| nucleofilament | yes | yes | yes | n.d. |
| DNA-bridging | yes | yes | yes | yes |
| DNA-bending | no | yes | no | yes |
| heteromers | yes | yes | predicted | predicted |
| *modulators* | | | | |
| paralogues | StpA, Hfp, H-NS2, H-NS$_{R27}$, Sfh | MvaU, Pmr | predicted | unidentified |
| truncated derivatives | H-NST | unidentified | unidentified | sRok |
| non-related interaction partners | Hha, YdgT, gp5.5, Ocr, Arn | Mip | HU | DnaA |

For H-NS, Lsr2 and MvaT, the sensitivity of protein–DNA nucleofilaments or bridged complexes to environmental changes *in vitro* is in agreement with the proposed role in regulation of genes sensitive to changes in the environment. However, the involvement in the physiological response to changes remains to be established for Lsr2 and MvaT.

## 5.2. Binding partners/antagonists

Not only physico-chemical conditions regulate the DNA binding and gene regulation properties of the four proteins. To date, such binding partners (paralogues, truncated derivatives, and non-related modulators and inhibitors) have been primarily identified for H-NS, but some modulators were also identified for MvaT, Lsr2 and Rok. They are summarized in table 1, together with the characteristics of the four proteins.

### 5.2.1. Paralogues

The H-NS paralogue StpA shares 58% sequence identity. Expression of StpA partially complements an *hns*⁻ phenotype of *E. coli* [104,105]. H-NS and StpA show negative autoregulation and repress transcription of each other's genes [106,107]. StpA is upregulated during growth at elevated temperature and high osmolarity [108,109]. Similar to H-NS, StpA can form dimers and higher-order oligomers *in vitro* [110]. *In vivo*, it is believed to exist only as heteromer with H-NS as it is otherwise susceptible to Lon degradation [111,112]. The structural effects of StpA binding to DNA *in vitro* are rather similar to H-NS: StpA can bridge DNA and forms protein filaments along DNA [33,113]. Despite the large similarities between the two proteins, H-NS, StpA and H-NS-StpA heteromers exhibit functionally distinct behaviour. At 20°C H-NS mediated DNA–DNA bridges induce transcriptional pausing, whereas they do not at 37°C (see § 5.1) [88]. The upregulation of StpA during growth at higher temperature might be explained by H-NS being unable to repress StpA at this temperature. At both 20 and 37°C, StpA does increase the pausing of RNAP, thereby repressing transcription [89]. StpA filaments on DNA are mostly present in a bridged conformation [89]. Also, bridged DNA–DNA complexes built using StpA-H-NS

heteromers are capable of inducing RNAP pausing at 37°C [89]. This robustness of StpA could contribute to gene silencing under stress conditions, forming an extra layer of gene regulation by H-NS. The presence of a third H-NS paralogue has been reported for several strains: the uropathogenic *E. coli* strain 536 expresses Hfp (also called H-NSB [114]) which is primarily expressed at 25°C and could thus be specifically implicated in regulating gene expression outside the host. [109] Heteromerization of Hfp with H-NS occurs, but, different from StpA this is not required for the stability of the protein [109]. The enteroaggregative *E. coli* strain 042 and several other *Enterobacteriaceae* also carry a second H-NS gene (or third in, e.g. strain 536 besides H-NS and Hfp) which can partially complement the *hns*⁻ phenotype [115]. This H-NS2 is expressed at significantly lower levels than H-NS during exponential growth, but the expression increases in the stationary phase [109]. Also, expression of H-NS2 is higher at 37°C than at 25°C, which could relate to the pathogenic nature of the investigated *E. coli* strains [109]. When comparing the amino acid sequences of all above mentioned H-NS paralogues, it becomes apparent that the N-terminal and C-terminal domain are quite conserved, particularly the DNA-binding domain (figure 5a). Because of this, the charge distribution of these proteins is also conserved, which makes it likely that they are similarly regulated by environmental conditions (electronic supplementary material). The differences in expression levels and thermal stability are most likely responsible for their different regulons.

Several plasmids have also been reported to encode H-NS paralogues. Often these proteins have distinct functional properties. The conjugative IncHI1 plasmid pSfR27 encodes the H-NS paralogue Sfh [116,117]. It was proposed that the plasmid moves from one host to another without causing a large change in host gene expression, because the binding of Sfh to pSfR27 prevents H-NS being titrated away from the chromosome [118]. Sfh can form homodimers and heterodimers with H-NS and StpA *in vivo*, but the DNA-binding properties of these complexes are still unknown [117]. The IncHI plasmid R27 encodes an H-NS variant that can partially complement the *hns*⁻ phenotype [119]. This H-NS$_{R27}$ binds to horizontally acquired DNA, but not to the core

royalsocietypublishing.org/journal/rsob    Open Biol. 9: 190223

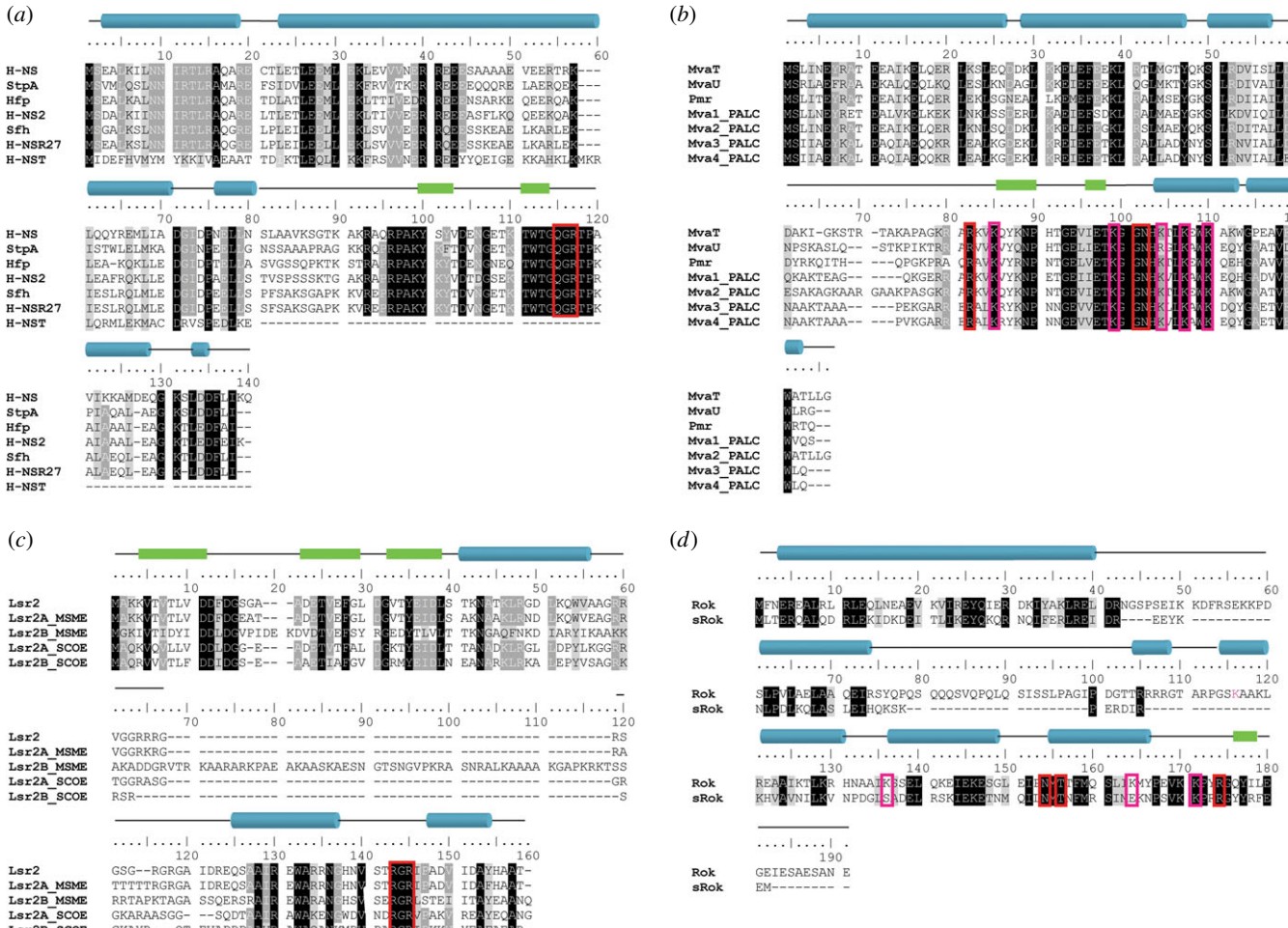

**Figure 5.** Comparison of H-NS-like architectural proteins and their structurally related protein–protein interaction partners. Alignment of identified paralogues and truncated derivatives of (a) H-NS, (b) MvaT, (c) Lsr2 and (d) Rok. Identical residues in all sequences are highlighted in black, identical residues in more than 75% of the sequences are highlighted in dark grey and conserved residues are highlighted in light grey. The DNA-binding motif is highlighted in red and the lysine network of MvaT and Rok in magenta. α-Helices are indicated with blue cylinders and β-sheets with green boxes. The indicated structure corresponds to the most upper sequence. H-NS, *E. coli* strain K-12, NP_415753.1; StpA, *E. coli* strain K-12, NP_417155.1; Hfp, *E. coli* strain 536, ABG69928.1; H-NS2, *E. coli* strain 042, CBG35667.1; Sfh, *Shigella flexneri* 2a, AAN38840.1; H-NSR27, *Salmonella enterica* subsp. *enterica* serovar Typhi strain CT18, NP_569380.1; H-NST, *E. coli* CFT073, NP_754305; MvaT, *P. aeruginosa* PAO1, NP_253005.1; MvaU, *P. aeruginosa* CLJ1, PTC37345.1; Pmr, *P. resinovorans*, NP_758612.1; Mva1/2/3/4_PALC, *P. alcaligenes* RU36E SIQ98833.1, SIQ72658.1, SIP93681.1 and SIP94365.1; Lsr2_MTUB, *M. tuberculosis* H37Rv, NP_218114.1; Lsr2A/B_MSMEG, *M. smegmatis* MKD8, AWT56911.1 and AWT52048.1; Lsr2A/B_SCOEL, *S. coelicolor* A(3)2, CAB40875.1 and CAB56356.1; Rok, *B. subtilis* strain 168, NP_389307.1; sRok, *B. subtilis* subsp. *natto*, YP_004243533.1.

genes regulated by chromosomal H-NS [119]. The difference is encoded in the linker region, which is hypothesized to be more rigid in H-NS$_{R27}$ [120]. Both Sfh and H-NS$_{R27}$ are quite conserved in N- and C-terminal domain compared with H-NS, and also the charge distribution is conserved (figure 5a; electronic supplementary material), which probably accounts for their 'stealth' function. These plasmid-borne H-NS homologues help the plasmid to stay unnoted by the host cell upon entry. By silencing horizontally acquired DNA (including the plasmid itself), they prevent H-NS from relieving its chromosomal targets and binding to the plasmid. This allows the plasmid to be transmitted without fitness costs for the host.

In *P. aeruginosa*, MvaU was identified as a paralogue of MvaT. The two proteins occupy the same chromosomal regions and work coordinately [35]. They are able to form heteromers, but this is not necessary for gene regulation [35,121]. Under conditions of fast growth, they are functionally redundant, becoming essential only when the other partner is deleted from the chromosome [35]. Uncharacterized paralogues of MvaT/U are present in several *Pseudomonas* genomes.

For example, in *P. alcaligenes* strain RU36E, four MvaT/U proteins are present (figure 5b). They share residues in the N-terminal domain and the AT-pincer motif. One of these is closely related to MvaT (Mva2: 81.7% sequence identity). The homology for the other three proteins is not that clear as the sequence identities to MvaT and MvaU are comparable (1: 54.7% and 52.1%; 3: 54.0% and 53.7%; 4: 53.2% and 54.1%). Another paralogue was found on the IncP-7 plasmid pCAR1 in *P. putida* [122]. The protein Pmr (plasmid-encoded MvaT-like regulator) shares 58% sequence identity with MvaT and *in vitro* forms homodimers and heteromers with MvaT-like proteins [123]. This is not surprising as many residues and the charge distribution are conserved between Pmr, MvaT and MvaU (figure 5b; electronic supplementary material). Pmr is able to regulate genes both on the plasmid and on the host chromosome [124]. Whereas the expression levels of the chromosomally encoded MvaT-like proteins alter between log and stationary phase, the level of Pmr is constant during different growth phases [125]. The regions on the chromosomes bound by Pmr and MvaT are identical, but their regulons differ [124]. This may be attributed to the formation

of different heteromers between Pmr and the two MvaT-like proteins with slightly different functions. Therefore, it remains unsure if Pmr is able to complement a *mvaT* phenotype in *P. putida*.

For Lsr2, several candidate paralogues have been identified so far. Several genomes of *Mycobacterium* species harbour two genes encoding for Lsr2 (e.g. *Mycobacterium smegmatis*) or carry a plasmid with an Lsr2 gene (e.g. *Mycobacterium gilvum*) [43]. In the case of *M. smegmatis* MKD8, some features of Lsr2, like its DNA-binding motif, are conserved between the two proteins, but the second Lsr2 has a much longer linker domain (figure 5*c*). It enlarges the positive patch already present in the Lsr2 linker, so it may provide the protein with extra charged surface for interdomain interactions (electronic supplementary material). It was noted that all *Streptomycetes* carry a second Lsr2 [126], although a recent study shows that Lsr2 is more important than the second, Lsr2-like protein [127]. They appear to be similar in charge distribution (electronic supplementary material). The DNA-binding modes of these candidate paralogues or the formation of heteromers with Lsr2 has not been investigated.

### 5.2.2. Truncated derivatives

Some pathogenic *E. coli* strains, like uropathogenic *E. coli* strain CFT073, encode a truncated version of H-NS: H-NST [114]. It lacks the DNA-binding domain and the oligomerization site (figure 5*a*). H-NST is able to counteract gene silencing by H-NS, which was proposed to occur by interfering with its oligomerization [114]. This model is supported by truncated H-NS products, mimicking H-NST, that are also able to perturb DNA bridging [61].

A small Rok variant (sRok) was identified on the *B. subtilis* plasmid pLS20 [128]. sRok can complement the *rok* phenotype in the competence pathway and associates genome-wide with the host chromosome. The DNA-binding motif in the C-terminal domain and parts of the N-terminal domain are conserved, suggesting that the function of these domains is also conserved (figure 5*d*). The difference between the two proteins is mostly the length of the linker. The neutral Q-linker of Rok is absent in sRok, which could lead to differences in DNA binding and responsiveness to changes in environmental conditions (figure 5*d*; electronic supplementary material). In contrast with *B. subtilis* that carries the *srok* gene nearly always on a plasmid, the *srok* gene is present on the chromosome of *B. licheniformis* and *B. paralicheniformis*. Possibly, the gene was transferred to the chromosome from the pLS20 plasmid. The *rok gene* is not present in all *Bacillus* species and its introduction has been attributed to a horizontal gene transfer event. [92]. The absence of *rok* in several *Bacillus* species means that its proposed genome organizing function is redundant and can be compensated for by other proteins. So far, it is unknown if Rok and sRok can form heteromers and how that would change their function in gene silencing and genome organization.

### 5.2.3. Non-related modulators and inhibitors

Hha and YdgT are members of the Hha/YmoA family acting as modulators of H-NS activity and function [129]. Different from H-NS truncated derivatives, these proteins have very limited sequence identity with H-NS. Hha is involved in co-regulation of a subset of known H-NS regulated genes,

especially in the silencing of horizontally acquired genes [119,130,131]. It is therefore not obvious *a priori* how these proteins would modulate the DNA-binding properties of H-NS. Hha was found to interact with the N-terminal domain of H-NS, specifically with the first two helices [132,133]. The H-NS-Hha co-crystal structure shows two Hha monomers binding to either site of the H-NS dimer, exposing two positively charged Hha surfaces per H-NS dimer [134]. Based on this structure, it was proposed that Hha affects H-NS mediated DNA bridging and thus coregulates specific genes with H-NS [134]. Indeed, it was shown that both Hha and YdgT enhance DNA bridging by H-NS [61]. Mechanistically, this could be explained in two ways: (i) Hha provides additional electrostatic interactions with DNA [134] or (ii) the H-NS dimer 'opens' upon Hha binding, resulting in a conformation capable of DNA bridging [61]. In the latter scenario, Hha could additionally stabilize the complex by the interactions implied in the first scenario. Hha has also been shown to enhance pausing of RNAP by H-NS and H-NS : Hha complexes preferentially bridge DNA [89]. In this manner, Hha could help with silencing a subset of H-NS regulated genes. Genomes of a wide range of pathogenic *E. coli* strains also contain two extra *hha* genes: *hha2* and *hha3* [135] in addition to the H-NS paralogues as described above. The presence of the extra *hha* genes is correlated to duplication of gene clusters that are regulated by H-NS and Hha and may be important for virulence [136]. Also the *hha2* gene is present in one of the duplicated regions, meaning that it originates from a duplication event. This shows a relation between duplicated virulence regions and extra genes encoding for regulators like H-NS and Hha. A gene encoding for Hha is also present on several plasmids, including the R27 plasmid described above [137,138].

H-NS can be inhibited by several phage-encoded proteins to counteract gene silencing. Gp5.5 from phage T7 interacts for example with the oligomerization domain of H-NS [139,140], thereby inhibiting oligomerization and gene silencing by H-NS. Another strategy used for counteracting gene silencing by H-NS is mimicking DNA, thereby competing with H-NS' genomic targets [141–143]. This strategy is used by Ocr from phage T7 and Arn from phage T4 [141–143]. Both strategies result in relieve of gene silencing by H-NS and making H-NS unable to bind to the genetic material of the phage. Lack of repression could lead to replication of the phage and entering of the lytic cycle to kill the host cell. Also MvaT can be inhibited by a protein encoded by a phage. The phage LUZ24 in *P. aeruginosa* expresses a protein called Mip (MvaT inhibiting protein), which was shown to perturb DNA binding of MvaT and proposed to inhibit the silencing of virus genes by MvaT [144].

Lsr2 of *M. tuberculosis* binds to the architectural protein HU [145]. This interaction involves the N-terminal domain of Lsr2 and the C-terminal tail of HU, which has (P)AKKA repeat motifs and thereby resembles histone tails. This tail is absent in HU of other bacteria discussed in this review (*E. coli*, *B. subtilis* and *Pseudomonas* species). The Lsr2–HU complex binds DNA, creating thick linear filaments instead of DNA bridges as seen for Lsr2 alone, or DNA compaction as seen for HU [145].

Rok interacts with bacterial replication initiator and transcription factor DnaA, and the two proteins jointly interact with a subset of Rok-bound genes [146]. DnaA enhances gene repression by Rok for this subset of Rok-bound genes.

royalsocietypublishing.org/journal/rsob    Open Biol. 9: 190223

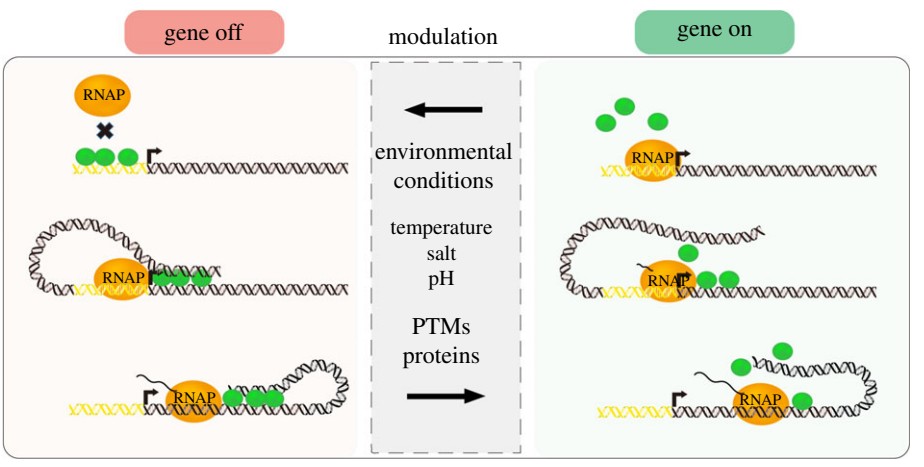

**Figure 6.** H-NS-like proteins functional regulation in gene silencing by different factors. The activity of RNAP transcription may be inhibited by H-NS-like proteins in three ways: (i) RNAP binding to promotor region can be inhibited by protein–DNA filaments or bridge complexes; (ii) RNAP elongation can be trapped and inhibited by DNA–protein–DNA bridge complexes; and (ii) elongating RNAP can be paused by DNA–protein–DNA complexes. The inhibition of RNAP by H-NS-like proteins may be modulated by factors such as environmental conditions (temperature, salt and pH), proteins and PTMs, allowing genes to be expressed.

DnaA binding to these genes is dependent on Rok and the DNA-binding domain of DnaA is neither necessary for this interaction, nor does it bind DNA in the Rok : DnaA complex. This is consistent with a model in which DnaA modulates the function of Rok. Because DnaA is an ATPase and its activity levels change during the cell cycle [147,148], it may be an indirect way to modulate the activity of Rok according to the cell cycle or energy status of the cell.

## 5.3. Post-translational modifications

The post-translational modification (PTMs) of eukaryotic histones has been studied for years and the functional significance of these modifications in biological processes including DNA repair, gene regulation and cell division is well established [149]. Although PTMs have been identified in recent years on NAPs in bacteria, their functional importance remains unclear.

H-NS has been found to undergo many PTMs which potentially add an extra layer to its function in chromatin organization and regulation [150]. PTMs discovered for H-NS include acetylation and succinylation of lysines, methylation of arginines, phosphorylation of serines, tyrosines and threonines, deamidation of asparagines and oxidation of methionines [150]. These modifications may influence diverse functional properties of H-NS such as DNA binding, oligomerization and interaction with other proteins. Acetylation or succinylation occurs on Lys96 and Lys121 near to the H-NS DNA-binding motif [151]. These modifications could reduce the DNA-binding affinity due to the change from positive to neutral or negative charge [152,153]. Acetylation at Lys83 and Lys87 located in the linker region has been identified [152,153]. This could decrease the H-NS DNA-binding affinity since the positive charge residues of the linker region are important for the DNA binding [55]. The identified phosphorylation on Tyr61 induces a negative charge and could interfere with H-NS oligomerization [154]. Indeed, the phosphorylation-mimicking mutation Y61D was shown to be important in H-NS dimer–dimer interaction. Lys6 has been shown to be involved in the interaction between Hha and H-NS [134]. Succinylation of Lys6 may reduce the strength of Hha binding by inducing steric hindrance, which could act in

regulation silencing of genes by H-NS at which Hha is involved as a co-partner [153].

Lsr2 of *M. tuberculosis* has been found to be phosphorylated at Thr112, located near the DNA-binding motif [155]. A recent study shows that phosphorylation of this residue decreases DNA binding by Lsr2, thus resulting in altered expression of genes important for *M. tuberculosis* growth and survival [156]. Some residues (Thr8, Thr22 and Thr31) in the N-terminal domain of Lsr2 were also found to be phosphorylated *in vitro* [156]. All these residues are located at β-sheets which are important for the formation of dimers or oligomers [43]. The PTMs on these residues could influence the dimerization or oligomerization of Lsr2 by the addition of negative charges. Also the interaction with HU, which involves the N-terminal domain of Lsr2 [145], could be regulated by phosphorylation.

In MvaT from *P. aeruginosa PA01*, several residues located at the N-terminal domain and linker region are acetylated and succinylated and at the C-terminal domain succinylated [157]. MvaT Lys86 is directly adjacent to Lys85, which is involved in DNA binding [46]. Lys86 is succinylated, leading to charge inversion and thereby a decrease in DNA-binding affinity [157]. Acetylation occurs on Lys22 and Lys31 located in the dimerization domain. On Lys39, shown to be important for oligomerization by forming hydrogen bonds [42], both acetylation and succinylation can occur. By altering positive charge to neutral or negative charge, these PTMs could affect the dimerization or oligomerization of MvaT. In addition, phosphorylation occurs at S2 in *P. putida PNL-MK25* MvaT, which may also affect its dimerization properties [158]. The linker region has been found to be important for H-NS interdomain interaction between C-terminal and N-terminal domain, which plays a role in H-NS controlling genes sensitive to temperature [64]. For the MvaT linker region, acetylation of Lys63 and succinylation of Lys72 was discovered. These PTMs, by changing positive charge to neutral or negative charge, could interfere with the intradomain interaction and modulate the function of MvaT in gene regulation.

Proteome analyses of *B. amyloliquefaciens* and *B. subtilis* show that Rok can be acetylated on residues K51 and K142 [159,160]. K51 is present in an (predicted) unstructured part of the N-terminal domain. Because it is currently unknown

which residues are important for dimerization and oligomerization of Rok, we cannot predict what the effect of acetylation on K51 would be. K142 is close to the DNA-binding motif of Rok and, while it was not identified as being directly involved, could affect the strength of the 'lysine network' [44]. Nevertheless, it could have an effect on the DNA-binding affinity of Rok by changing the charge of K142 from positive to neutral.

PTMs that lead to changes in charge could affect the charge distribution of H-NS-like proteins. This may alter the proteins' response to environmental changes, which is important for their role in gene regulation. Functional studies on PTMs on these proteins are currently lacking, but they will prove essential in better understanding the function of PTMs in global gene regulation and physiological adaptation.

# 6. Conclusion and perspectives

The architectural chromatin proteins H-NS, MvaT, Lsr2 and the newly proposed functional homologue Rok play important roles in the organization and regulation of the bacterial genome. Although their sequence similarity is low, their domain organization is the same: the N-terminal domain functions in dimerization and oligomerization, the C-terminal domain binds DNA and the two are connected by a flexible linker. Except for Rok, the charge distribution of these proteins along the sequence is highly conserved.

The four proteins are all capable of bridging DNA duplexes, which is important for both spatial genome organization and gene regulation (figure 6). Although a vast amount of research is done on gene regulation by H-NS, it remains to be investigated if the other three proteins regulate genes in similar ways. To decipher mechanistically how H-NS represses genes, more advanced single-molecule and *in vivo* experiments are needed.

H-NS, MvaT and Lsr2 are functionally modulated by changes in environmental conditions, protein partners and PTMs (figure 6). PTMs are to date poorly explored and could be a completely new field of research. The first functional study for phosphorylation of Lsr2 shows that PTMs can indeed be important for the DNA-binding properties of these chromatin organizing proteins and could be a new, uncharacterized way of regulation.

We propose that the shared domain organization and asymmetric charge distribution of the H-NS-like proteins is key to their response to changes in environmental conditions. This information could be used to predict a protein's behaviour and may be employed in fighting pathogenic strains by either activating or repressing specific gene transcription.

Data accessibility. This article has no additional data.

Authors' contributions. The article was jointly written by Q.L. and A.M.E. under supervision of R.T.D. F.B.B. performed protein structural analysis and contributed to writing. All authors read and approved of the manuscript.

Competing interests. We declare we have no competing interests.

Funding. Research on the topic of this review in the laboratory of R.T.D. is supported by a grant from the Netherlands Organization for Scientific Research (grant no. VICI 016.160.613).

Acknowledgements. We acknowledge Dr Wiep Klaas Smits for critical reading of the manuscript.

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
