## [Reviewer comments · Open Biology]

Review History

RSOB-19-0223.R0 (Original submission)

Review form: Reviewer 1

Recommendation

Accept with minor revision (please list in comments)

Do you have any ethical concerns with this paper?

No

Comments to the Author

This submission from Qin and coworkers nicely summarizes the current understanding of the DNA binding property of H-NS-like proteins. The submission is reasonably well written from that point of view, but some issues should be addressed:

The references should be cited much more carefully. A few examples are listed below.

Lines 237 – 240: “Evidence in support of formation of nucleoprotein filaments comes from single-molecule studies... (37, 54, 55).” The reference #37 is not a single-molecule study. The first single-

molecule study showing DNA stiffening by H-NS was reported by Amit et al. (Ref. 30). The second single-molecule study was reported by Liu et al. (Ref. 59), which also revealed that divalent cations Mg^{2+} and Ca^{2+} could switch the H-NS from DNA stiffening to DNA binding. The third single-molecule study was reported by Lim et al., showing that StpA forms filament first and that filament could subsequently mediate DNA bridging (Ref. 111). These references and later studies on MvaT, Lsr2, MvaU and H-NS binding to supercoiled DNA should be cited in chronicle order to highlight the original findings.

Lines 237 – 240: “Evidence in support of formation of nucleoprotein filaments comes from single-molecule studies... (37, 54, 55).” The sentence is inaccurate since in most of the above-mentioned studies, the filament formation was also visualized with AFM.

Lines 259-260: “The switch between the two DNA binding modes involves a conformational change of the H-NS dimers from a half-open to an open conformation driven by Mg^{2+} ”. The work by Liu et al. (Ref. 59) should be cited here, since it was the first study reporting the Mg^{2+} and Ca^{2+} mediated switch from the pure stiffening filament to DNA bridging. Such switching was also observed for several other H-NS like proteins such as Lsr2 and MvaT in later single-molecule studies, which should also be cited in chronicle order.

Lines 232-233: “3). First, the C-terminal DNA-binding domain directs the protein to a high-affinity site (nucleation) (52).” This view neglects the important role of the linker in the DNA binding by H-NS. In a recent work published by Gao et al. (PNAS 2017, 114 (47) 12560-12565), it was found that the charged residues in the linker play a critical role in the DNA-binding by H-NS. It is reasonable to assume that the positively charged residues in the linker recruits H-NS to DNA non-specifically. This then allows H-NS to scan on DNA to search for the specific site where the C-terminal domain can engage with a higher affinity. Such a sliding-diffusion mechanism has been well known to drastically speed to the target searching process of DNA binding proteins (e.g., a nice review by Halford and Marko, Nucleic Acids Research, 2004, 32: 3040–3052). The differential roles of the linker and the C-terminal domain in DNA binding affinity and specificity were fully quantified in a more recent single-molecule study (Gulvady et al., 2018, Nucleic acids research 46 (19), 10216-10224). These recent progresses should be discussed in the review and the relevant works should be acknowledged.

Review form: Reviewer 2

Recommendation

Accept with minor revision (please list in comments)

Do you have any ethical concerns with this paper?

No

Comments to the Author

This manuscript provides a comprehensive and timely review on bacterial nucleoid-associated proteins, focusing specifically on the DNA-bridging proteins H-NS, MvaT, Lsr2 and Rok. The coverage of the subject is exhaustive and detailed, as also evidenced by the inclusion of 159 articles in the reference list. The introduction puts the selected proteins into the broader context of architectural factors that are involved in chromatin organization and DNA compaction. Then the Authors examine in great depth different aspects and features of the specific DNA-bridging proteins, ranging from structural details to functional properties. The manuscript layout and flow of information are both excellent. The narrative is very smooth and the style makes the reading

very easy and enjoyable. Given that chromosome topology and organization is a hot topic right now, this review will be an excellent resource for scientists who work on genome biology, DNA-protein interactions and DNA-binding proteins. The manuscript is well written and I have only some minor suggestions that are listed below:

1. page 16, if space allows it, the Authors could include a figure that shows the mechanism through which H-NS-like proteins switch between the two binding modes described in that section;
2. page 22, line 372, replace 'implied' with 'involved' or 'implicated';
3. page 23, line 395, is it possible that other proteins different from Rok could control genes in a temperature-based fashion? The statement '...suggesting that genes in *B. subtilis* are not regulated by temperature' sounds very definitive and final;
4. page 26, line 464, I would suggest to change the sentence to 'At 20°C H-NS mediated DNA-DNA bridges induce transcriptional pausing, whereas they do not at 37°C';
5. page 28, line 494, change the sentence to '...because the binding of Sfh to pSfr27 prevents H-NS being titrated away from the chromosome';
6. page 30, line 549, replace 'truncates' with 'truncated H-NS products';
7. page 33, line 602, modify the sentence to 'Lack of repression could lead to replication of the phage...';
8. page 33, line 607, I would suggest to change the sentence to 'Lsr2 of *M. tuberculosis* binds to the HU architectural protein';
9. figure 5, the DNA-binding motif highlighted in red is not very visible, I had to look very closely to spot the red; maybe the motif could be highlighted differently;
10. page 37, line 667, change 'hinderance' to 'hindrance';
11. page 38, line 681, change 'implied' to 'involved';
12. page 39, line 704, change the sentence to 'This may alter the proteins' response to environmental changes...';
13. page 39, line 724, change sentence to 'PTMs are to date poorly explored...'

Decision letter (RSOB-19-0223.R0)

04-Nov-2019

Dear Dr Dame

We are pleased to inform you that your manuscript RSOB-19-0223 entitled "The architects of bacterial DNA bridges: a structurally and functionally conserved family of proteins" has been accepted by the Editor for publication in *Open Biology*. The reviewer(s) have recommended publication, but also suggest some minor revisions to your manuscript. Therefore, we invite you to respond to the reviewer(s)' comments and revise your manuscript.

Please submit the revised version of your manuscript within 7 days. If you do not think you will be able to meet this date please let us know immediately and we can extend this deadline for you.

- 1) A text file of the manuscript (doc, txt, rtf or tex), including the references, tables (including captions) and figure captions. Please remove any tracked changes from the text before submission. PDF files are not an accepted format for the "Main Document".
- 2) A separate electronic file of each figure (tiff, EPS or print-quality PDF preferred). The format should be produced directly from original creation package, or original software format. Please note that PowerPoint files are not accepted.
- 3) Electronic supplementary material: this should be contained in a separate file from the main text and meet our ESM criteria (see <http://royalsocietypublishing.org/instructions-authors#question5>). All supplementary materials accompanying an accepted article will be treated as in their final form. They will be published alongside the paper on the journal website and posted on the online figshare repository. Files on figshare will be made available approximately one week before the accompanying article so that the supplementary material can be attributed a unique DOI.

Online supplementary material will also carry the title and description provided during submission, so please ensure these are accurate and informative. Note that the Royal Society will not edit or typeset supplementary material and it will be hosted as provided. Please ensure that the supplementary material includes the paper details (authors, title, journal name, article DOI). Your article DOI will be 10.1098/rsob.2016[last 4 digits of e.g. 10.1098/rsob.20160049].

- 4) A media summary: a short non-technical summary (up to 100 words) of the key findings/importance of your manuscript. Please try to write in simple English, avoid jargon, explain the importance of the topic, outline the main implications and describe why this topic is newsworthy.

Images

Data-Sharing

It is a condition of publication that data supporting your paper are made available. Data should be made available either in the electronic supplementary material or through an appropriate repository. Details of how to access data should be included in your paper. Please see <http://royalsocietypublishing.org/site/authors/policy.xhtml#question6> for more details.

Data accessibility section

Sincerely,
The Open Biology Team
mailto:openbiology@royalsociety.org

Reviewer(s)' Comments to Author:

Referee: 1

Comments to the Author(s)

This submission from Qin and coworkers nicely summarizes the current understanding of the DNA binding property of H-NS-like proteins. The submission is reasonably well written from that point of view, but some issues should be addressed:

The references should be cited much more carefully. A few examples are listed below.

Lines 237 - 240: "Evidence in support of formation of nucleoprotein filaments comes from single-molecule studies... (37, 54, 55)." The reference #37 is not a single-molecule study. The first single-molecule study showing DNA stiffening by H-NS was reported by Amit et al. (Ref. 30). The second single-molecule study was reported by Liu et al. (Ref. 59), which also revealed that divalent cations Mg²⁺ and Ca²⁺ could switch the H-NS from DNA stiffening to DNA binding. The third single-molecule study was reported by Lim et al., showing that StpA forms filament first and that filament could subsequently mediate DNA bridging (Ref. 111). These references and later studies on MvaT, Lsr2, MvaU and H-NS binding to supercoiled DNA should be cited in chronicle order to highlight the original findings.

Lines 237 - 240: "Evidence in support of formation of nucleoprotein filaments comes from single-molecule studies... (37, 54, 55)." The sentence is inaccurate since in most of the above-mentioned studies, the filament formation was also visualized with AFM.

Lines 259-260: "The switch between the two DNA binding modes involves a conformational change of the H-NS dimers from a half-open to an open conformation driven by Mg²⁺". The work by Liu et al. (Ref. 59) should be cited here, since it was the first study reporting the Mg²⁺ and Ca²⁺ mediated switch from the pure stiffening filament to DNA bridging. Such switching was also observed for several other H-NS like proteins such as Lsr2 and MvaT in later single-molecule studies, which should also be cited in chronicle order.

Lines 232-233: "(3). First, the C-terminal DNA-binding domain directs the protein to a high-affinity site (nucleation) (52)." This view neglects the important role of the linker in the DNA binding by H-NS. In a recent work published by Gao et al. (PNAS 2017, 114 (47) 12560-12565), it

was found that the charged residues in the linker play a critical role in the DNA-binding by H-NS. It is reasonable to assume that the positively charged residues in the linker recruits H-NS to DNA non-specifically. This then allows H-NS to scan on DNA to search for the specific site where the C-terminal domain can engage with a higher affinity. Such a sliding-diffusion mechanism has been well known to drastically speed to the target searching process of DNA binding proteins (e.g., a nice review by Halford and Marko, *Nucleic Acids Research*, 2004, 32: 3040–3052). The differential roles of the linker and the C-terminal domain in DNA binding affinity and specificity were fully quantified in a more recent single-molecule study (Gulvady et al., 2018, *Nucleic acids research* 46 (19), 10216-10224). These recent progresses should be discussed in the review and the relevant works should be acknowledged.

Referee: 2

Comments to the Author(s)

This manuscript provides a comprehensive and timely review on bacterial nucleoid-associated proteins, focusing specifically on the DNA-bridging proteins H-NS, MvaT, Lsr2 and Rok. The coverage of the subject is exhaustive and detailed, as also evidenced by the inclusion of 159 articles in the reference list. The introduction puts the selected proteins into the broader context of architectural factors that are involved in chromatin organization and DNA compaction. Then the Authors examine in great depth different aspects and features of the specific DNA-bridging proteins, ranging from structural details to functional properties. The manuscript layout and flow of information are both excellent. The narrative is very smooth and the style makes the reading very easy and enjoyable. Given that chromosome topology and organization is a hot topic right now, this review will be an excellent resource for scientists who work on genome biology, DNA-protein interactions and DNA-binding proteins. The manuscript is well written and I have only some minor suggestions that are listed below:

1. page 16, if space allows it, the Authors could include a figure that shows the mechanism through which H-NS-like proteins switch between the two binding modes described in that section;
2. page 22, line 372, replace 'implied' with 'involved' or 'implicated';
3. page 23, line 395, is it possible that other proteins different from Rok could control genes in a temperature-based fashion? The statement '...suggesting that genes in *B. subtilis* are not regulated by temperature' sounds very definitive and final;
4. page 26, line 464, I would suggest to change the sentence to 'At 20°C H-NS mediated DNA-DNA bridges induce transcriptional pausing, whereas they do not at 37°C';
5. page 28, line 494, change the sentence to '...because the binding of Sfh to pSfR27 prevents H-NS being titrated away from the chromosome';
6. page 30, line 549, replace 'truncates' with 'truncated H-NS products';
7. page 33, line 602, modify the sentence to 'Lack of repression could lead to replication of the phage...';
8. page 33, line 607, I would suggest to change the sentence to 'Lsr2 of *M. tuberculosis* binds to the HU architectural protein';
9. figure 5, the DNA-binding motif highlighted in red is not very visible, I had to look very closely to spot the red; maybe the motif could be highlighted differently;
10. page 37, line 667, change 'hinderance' to 'hindrance';
11. page 38, line 681, change 'implied' to 'involved';
12. page 39, line 704, change the sentence to 'This may alter the proteins' response to environmental changes...';
13. page 39, line 724, change sentence to 'PTMs are to date poorly explored...'

Author's Response to Decision Letter for (RSOB-19-0223.R0)

See Appendix A.

Decision letter (RSOB-19-0223.R1)

08-Nov-2019

Dear Dr Dame

We are pleased to inform you that your manuscript entitled "The architects of bacterial DNA bridges: a structurally and functionally conserved family of proteins" has been accepted by the Editor for publication in Open Biology.

Sincerely,

The Open Biology Team
mailto:openbiology@royalsociety.org

Appendix A

Response to Reviewer Comments for Manuscript RSOB-19-0223.

Author responses are in blue text.

Referee: 1

Comments to the Author(s)

This submission from Qin and coworkers nicely summarizes the current understanding of the DNA binding property of H-NS-like proteins. The submission is reasonably well written from that point of view, but some issues should be addressed:

The references should be cited much more carefully. A few examples are listed below.

Lines 237 – 240: “Evidence in support of formation of nucleoprotein filaments comes from single-molecule studies... (37, 54, 55).” The reference #37 is not a single-molecule study. The first single-molecule study showing DNA stiffening by H-NS was reported by Amit et al. (Ref. 30). The second single-molecule study was reported by Liu et al. (Ref. 59), which also revealed that divalent cations Mg^{2+} and Ca^{2+} could switch the H-NS from DNA stiffening to DNA binding. The third single-molecule study was reported by Lim et al., showing that StpA forms filament first and that filament could subsequently mediate DNA bridging (Ref. 111). These references and later studies on MvaT, Lsr2, MvaU and H-NS binding to supercoiled DNA should be cited in chronicle order to highlight the original findings.

We agree with the reviewer and have added the suggested references to highlight the original findings.

Lines 237 – 240: “Evidence in support of formation of nucleoprotein filaments comes from single-molecule studies... (37, 54, 55).” The sentence is inaccurate since in most of the above-mentioned studies, the filament formation was also visualized with AFM.

We agree with the reviewer. Now the sentence is: “Evidence in support of formation of nucleoprotein filaments comes from atomic force microscopy (AFM) and single-molecule studies which revealed that H-NS, Lsr2 and MvaT all form rigid protein-DNA filaments, suggestive of protein oligomerization along DNA”

Lines 259-260: “The switch between the two DNA binding modes involves a conformational change of the H-NS dimers from a half-open to an open conformation driven by Mg^{2+} ”. The work by Liu et al. (Ref. 59) should be cited here, since it was the first study reporting the Mg^{2+} and Ca^{2+} mediated switch from the pure stiffening filament to DNA bridging. Such switching was also

observed for several other H-NS like proteins such as Lsr2 and MvaT in later single-molecule studies, which should also be cited in chronicle order.

We thank the reviewer's suggestion and have now cited the work by Liu et al.

Lines 232-233: "3). First, the C-terminal DNA-binding domain directs the protein to a high-affinity site (nucleation) (52)." This view neglects the important role of the linker in the DNA binding by H-NS. In a recent work published by Gao et al. (PNAS 2017, 114 (47) 12560-12565), it was found that the charged residues in the linker play a critical role in the DNA-binding by H-NS. It is reasonable to assume that the positively charged residues in the linker recruits H-NS to DNA non-specifically. This then allows H-NS to scan on DNA to search for the specific site where the C-terminal domain can engage with a higher affinity. Such a sliding-diffusion mechanism has been well known to drastically speed to the target searching process of DNA binding proteins (e.g., a nice review by Halford and Marko, Nucleic Acids Research, 2004, 32: 3040–3052). The differential roles of the linker and the C-terminal domain in DNA binding affinity and specificity were fully quantified in a more recent single-molecule study (Gulvady et al., 2018, Nucleic acids research 46 (19), 10216-10224). These recent progresses should be discussed in the review and the relevant works should be acknowledged.

We agree with the reviewer comments which remind us with the critical role of the linker region is the H-NS-DNA complex formation. This aspect is now plainly stated in the text and the relevant works are acknowledged.

"First, the C-terminal DNA-binding domain directs the protein to a high-affinity site (nucleation) (52). This step appears to be assisted by the positively charged amino acid residues of the linker region which interact with the DNA and recruit H-NS to bind non-specifically (54–56). This then allows H-NS to scan on DNA to search for the specific site where the C-terminal domain can engage with a higher affinity".

Referee: 2

Comments to the Author(s)

This manuscript provides a comprehensive and timely review on bacterial nucleoid-associated proteins, focusing specifically on the DNA-bridging proteins H-NS, MvaT, Lsr2 and Rok. The coverage of the subject is exhaustive and detailed, as also evidenced by the inclusion of 159 articles in the reference list. The introduction puts the selected proteins into the broader context

of architectural factors that are involved in chromatin organization and DNA compaction. Then the Authors examine in great depth different aspects and features of the specific DNA-bridging proteins, ranging from structural details to functional properties. The manuscript layout and flow of information are both excellent. The narrative is very smooth and the style makes the reading very easy and enjoyable. Given that chromosome topology and organization is a hot topic right now, this review will be an excellent resource for scientists who work on genome biology, DNA-protein interactions and DNA-binding proteins. The manuscript is well written and I have only some minor suggestions that are listed below:

1. page 16, if space allows it, the Authors could include a figure that shows the mechanism through which H-NS-like proteins switch between the two binding modes described in that section;

We thank the reviewer for the helpful comment. In figure 4 a schematic representation of the switching mechanism between H-NS proteins DNA binding modes is include in panel (a).

2. page 22, line 372, replace 'implied' with 'involved' or 'implicated';

We agree with the reviewer and have replaced 'implied' with 'involved'.

3. page 23, line 395, is it possible that other proteins different from Rok could control genes in a temperature-based fashion? The statement '...suggesting that genes in *B. subtilis* are not regulated by temperature' sounds very definitive and final;

We agree with the reviewer and have revised the statement to say: '...suggesting that genes in *B. subtilis* that are repressed by Rok are not regulated by temperature'

4. page 26, line 464, I would suggest to change the sentence to 'At 20°C H-NS mediated DNA-DNA bridges induce transcriptional pausing, whereas they do not at 37°C';

We appreciate the reviewer's suggestion and have changed the sentence to 'At 20°C H-NS mediated DNA-DNA bridges induce transcriptional pausing, whereas they do not at 37°C'.

5. page 28, line 494, change the sentence to '...because the binding of Sfh to pSfR27 prevents H-NS being titrated away from the chromosome';

We agree with the reviewer and have changed the sentence to '...because the binding of Sfh to pSfR27 prevents H-NS being titrated away from the chromosome'.

6. page 30, line 549, replace 'truncates' with 'truncated H-NS products';

We have made this change.

7. page 33, line 602, modify the sentence to 'Lack of repression could lead to replication of the phage...';

We agree with the reviewer and have modified the sentence to 'Lack of repression could lead to replication of the phage...'.

8. page 33, line 607, I would suggest to change the sentence to 'Lsr2 of M. tuberculosis binds to the HU architectural protein';

We thank the reviewer's suggestion and have changed the sentence to 'Lsr2 of M. tuberculosis binds to the architectural protein HU'.

9. figure 5, the DNA-binding motif highlighted in red is not very visible, I had to look very closely to spot the red; maybe the motif could be highlighted differently;

We appreciate the reviewer's comment and have updated the figure to see the DNA binding domain more clearly.

10. page 37, line 667, change 'hinderance' to 'hindrance';

We have made this change.

11. page 38, line 681, change 'implied' to 'involved';

We have made this change.

12. page 39, line 704, change the sentence to 'This may alter the proteins' response to environmental changes...';

We thank the reviewer's suggestion and have changed the sentence to 'This may alter the proteins' response to environmental changes...'.

13. page 39, line 724, change sentence to 'PTMs are to date poorly explored...'.

We thank the reviewer's suggestion and have changed the sentence to 'PTMs are to date poorly explored...'.